# Inequality indices to monitor geographic differences in incidence, mortality and fatality rates over time during the COVID-19 pandemic

**Kirsi M. Manz** *, **Ulrich Mansmann**

Institute of Medical Information Processing, Biometry and Epidemiology, Ludwig-Maximilians University Munich, Munich, Germany

* manz@ibe.med.uni-muenchen.de

**Data Availability Statement:** All data can be found in the OSF data repository (https://osf.io/c3ren/).

**Funding:** The authors received no specific funding for this work.

## Abstract

### Background

It is of interest to explore the variability in how the COVID-19 pandemic evolved geographically during the first twelve months. To this end, we apply inequality indices over regions to incidences, infection related mortality, and infection fatality rates. If avoiding of inequality in health is an important political goal, a metric must be implemented to track geographical inequality over time.

### Methods

The relative and absolute Gini index as well as the Theil index are used to quantify inequality. Data are taken from international data bases. Absolute counts are transformed to rates adjusted for population size.

### Results

Comparing continents, the absolute Gini index shows an unfavorable development in four continents since February 2020. In contrast, the relative Gini as well as the Theil index support the interpretation of less inequality between European countries compared to other continents. Infection fatality rates within the EU as well as within the U.S. express comparable improvement towards more equality (as measured by both Gini indices).

### Conclusions

The use of inequality indices to monitor changes in geographic inequality over time for key health indicators is a valuable tool to inform public health policies. The absolute and relative Gini index behave complementary and should be reported simultaneously in order to gain a meta-perspective on very complex dynamics.

**Competing interests:** The authors have declared that no competing interests exist.

## Introduction

It is of interest to evaluate the distributional inequality (between regions, over time) of key indicators representing COVID-19 effects: Incidence (rates, IR), mortality (rates, MR), and the infection fatality (rate, IFR). These measures inform global as well as regional health policies. Many sources, such as private, institutional, national, or international dashboards provide these information as maps or enumerated in long lists. The information presented reflect global geographic scenarios or are focused on smaller scale areas. How different is the documented pandemic activity worldwide, in particular between continents, between nations or between sub-regions of larger geographical entities? How does this heterogeneity change over time and how does it depend on geographic scale? Furthermore, it is of interest to study inequality across regions with a common health policy framework, such as the European Union (EU) or the United States of America (U.S.).

In general, such maps or listings are not accompanied by general purpose measures to quantify heterogeneity or inequality. Often, the analyst uses the "eyeball test" to scan over the data quickly. Sometimes, *caterpillar plots* represent estimates of the same parameter, such as incidence rate and illustrate their variability over a set of units. Following information presented in maps or long lists over time is a further challenge. However, a more formal approach is needed to study heterogeneity or inequality of complex dynamics over space and time.

In this paper, heterogeneity is studied in terms of inequality. Inequality measures examine the distribution of a single variable whose data are arranged in a monotonic order. A theoretical and methodological discussion of how to measure inequality is found in the economic literature [1].

Among the most common inequality metrics are the *relative Gini index* ($GI_{rel}$) and the *Theil index* (TI). The $GI_{rel}$ is the mean absolute distance between the observations divided by their mean. It takes values between 0 and 1 (0 equality, 1 maximal inequality). The reason for its popularity is that the $GI_{rel}$ is equal to the ratio of two areas in Lorenz curve diagrams, which are widely used in economics [2].

The *Theil index* is entropy based and decreases with increasing entropy. As for any distribution and with reference to information theory, *maximum entropy* occurs once the observations cannot be distinguished by their values, i.e. when there is perfect equality. The more "distinguishable" observations are, the lower is the "actual entropy" of a system consisting of regions with a specific disease status. The TI has obvious, intuitive, plausible and natural justification, rather than just being justified in terms of entropy. It's maximum is given by the natural logarithm of the population size.

This paper also studies the *absolute Gini index* ($GI_{abs}$), which is the mean absolute difference between observations. The $GI_{abs}$ is scale dependent. For X, the variable of interest, it holds that $GI_{rel}(c \cdot X) = GI_{rel}(X)$, but $GI_{abs}(c \cdot X) = c \cdot GI_{abs}(X)$. If absolute differences do not change but the mean of the variable will increase over time, the $GI_{abs}$ remains the same but the $GI_{rel}$ will decrease over time (see Eqs (1) and (2)). It is recommended to use the $GI_{abs}$ as the more reliable measure of inequality for time dependent analyses compared to $GI_{rel}$ [3].

It is not common to use inequality indices based on quadratic distances like variance or standard deviation. Metrics based on such squared distances diminish small (<1) and exaggerate larger differences (>1).

Inequality indices have been introduced by economists. However, besides their use in economic settings they also play a role in health-related topics, where they are mainly used to study *health equity*. Equity goals, such as equal treatment for equal need or equality of access to health care, are central to health policy [4]. Specific applications of the $GI_{rel}$ and $GI_{abs}$ regarding equity in health and health care are geographical inequality in mortality and life

expectancy, and access to and distribution of health care resources [5–11]. Inequality indices are also used to study the spread of infectious diseases. How does the distribution of sexually transmitted diseases depend on level of sexual activity [12]? How is the inequality between malaria prevalence among populations [13]? Inequality studies also concern the distribution of patient recruitment into clinical trials in different settings [14].

Williams and Doessel [15] review the economic literature on inequality measures from the perspective of health scientists. They stress the relevance of measures, such as Gini indices, Theil's Index of Entropy or Atkinson's Measure [1] and state: "*It is often important in measuring inequality to report several measures/indexes, within the constraints of the data available, and to examine the strengths and weaknesses of each measure. In so doing, the nature of the inequality is depicted more accurately, and one can weight equity judgements more wisely, than is possible by emphasising any single measure of inequality*".

Health equity goals for COVID-19 are formulated by several agencies, see for example [16]. Inequities in social determinants of health that put groups at increased risk of getting sick and dying from COVID-19 include: healthcare and utilization; occupation, educational, income, or wealth gaps; and housing. Achieving health equity requires as a first step quantifying health inequality before focusing efforts on preventable inequities and eliminating inequities in health and health care. The population health impact of COVID-19 has exposed longstanding inequities that have systematically undermined the physical, social, economic, and emotional health of racial and ethnic minority populations and other population groups that are bearing a disproportionate burden of COVID-19 [17].

To the best of our knowledge, inequality in COVID-19 infections and related deaths has not been quantitatively investigated by using indices of inequality. We also explore inequality in infection fatality rates. A specific focus of our paper is the comparison of inequality between the United States and the European Union.

## Materials and methods

### The data

Global infection and death data were retrieved from the Johns Hopkins University Center for Systems Science and Engineering (JHU CSSE) COVID-19 Data Repository [18]. Data between January 29, 2020 and February 2, 2021 were used. Population data for each country were extracted from the COVID-19 database of the European Centre for Disease Control and Prevention (ECDC) [19], which includes a population estimate for 2019. Data on the assignment of countries to continents were also extracted from the ECDC data. A subset of the JHU CSSE data was the basis for the analysis of the EU and the U.S. data. The list of the current member states of the EU was obtained from the official website of the EU [20] and excludes the United Kingdom. Of note, the continent *Europe* contains a broader selection of countries. Population data of 2019 for each U.S. state were obtained from the United States Census Bureau [21]. We did not include Puerto Rico in our analyses and also excluded all cruise ship data and data regarding repatriated travellers. Due to large geographic distances, we also excluded data from all overseas territories for the Netherlands, the United Kingdom, and France.

Note that because of the small number of countries and infections and deaths within countries in Oceania, measures of inequality were not calculated for this continent. The nine affected countries are Australia, New Zealand, Papua New Guinea, Fiji, Solomon Islands, Marshall Island, Samoa, Micronesia and Vanuatu. In the latter five countries, a total of 80 infections and no COVID-19 related deaths have been reported up to February 2, 2021.

The data were analyzed with R (Version 3.6.3) [22]. Both Gini indices were calculated using the DescTools package [23], the Theil index was evaluated using the dineq package [24] and for the likelihood-ratio test the package lmtest was used [25].

## The disease measures

The incidence and mortality parameters (IR(t) and MR(t)) at day t are the cumulative seven days incidences per 100,000 persons before day t. The data was aggregated over the past seven days and we are working with discrete time points of t = 7, 14, 21, etc. The IFR(t) at day t is the ratio of MR(t)/IR(t-14). The chosen time shift follows an established practice [26, 27]. As described above, we use population size data in combination with data on the corresponding positive test results and fatalities associated with COVID-19. There will be no correction for undetected SARS-CoV-2 infections and no adjustment for testing strategies. Formally we work with the *test-positive fatality ratio* (TPFR), which is the ratio of documented COVID-19 related deaths to documented positively tested persons. Strictly speaking, TPFR should be distinguished from IFR, which includes all infected persons (with and without symptoms) in the denominator. Because of untested infected persons without symptoms, it is plausible that IFR $\leq$ TPFR. Following common parlance, we use the term *naive IFR* (nIFR) instead of TPFR.

The disease measures are adjusted for population size that makes counts more comparable across countries. No adjustment for sex and age is feasible due to missing data. Time-course data of inequality indices between parts of a unit is based on counts aggregated within a part over non-overlapping short periods of seven days.

## Indices of inequality

The empirical *relative Gini index* is defined as

$$\text{GI}_{\text{rel}} = \frac{\sum_{i=1}^{n} \sum_{j=1}^{n} |x_i - x_j|}{2n^2 \bar{x}}, \tag{1}$$

where $x$ is an observed value (such as a specific incidence rate), $n$ is the number of values observed and $\bar{x}$ is the mean value. Alternative formulae can be found in [28]. The $\text{GI}_{\text{rel}}$ takes values between 0 and 1, which follows from the inequality $|x - y| \leq |x| + |y|$. The $\text{GI}_{\text{rel}}$ is scale invariant. In a setting of $n$ parts of a unit where $n - 1$ parts have value 0 and one part has value $w$, it follows that $\bar{x} = \frac{w}{n}$ and $\text{GI}_{\text{rel}} = \frac{2 \cdot (n-1) \cdot w \cdot n}{2 \cdot n^2 \cdot w} \sim 1$.

We combine this invariance with the nIFR: MR(t) = nIFR(t) · IR(t-14). Here MR(t) represents the incidence of death at day t given the nIFR(t) and IR(t-14), the naive IFR at day t and the incidence of infected 14 days before. Assuming a constant nIFR(t) over the parts of a larger unit, it holds $\text{GI}_{\text{rel,Infection}} = \text{GI}_{\text{rel,Death}}$.

There are no established cut-offs to qualify specific $\text{GI}_{\text{rel}}$ values. The presented figures for the $\text{GI}_{\text{rel}}$ show quintile regions defined by 0.2, 0.4, 0.6, 0.8, and 1. This allows to have a neutral category in the middle and two ratings for inequality (strong: 0.6 to 0.8; high: 0.8 to 1) as well as equality (in terms of inequality low: 0.2 to 0.4; diminished; 0 to 0.2). The $\text{GI}_{\text{rel}}$ of exponential distribution with rate 1 is about 0.5 and of uniform distribution about 0.33. A gamma distribution with mean 20 and variance 40 has a $\text{GI}_{\text{rel}}$ of 0.18. The $\text{GI}_{\text{rel}}$ of a Pareto distribution with $\alpha$ = 0.7 and threshold equal to 100 is close to 1.

The empirical *absolute Gini index* can be calculated as

$$\text{GI}_{\text{abs}} = \frac{\sum_{i=1}^{n} \sum_{j=1}^{n} |x_i - x_j|}{2n^2}, \tag{2}$$

where $x$ is an observed value (such as a specific incidence rate) and $n$ is the number of values observed. In a setting of $n$ parts of a unit where $n − 1$ parts have value 0 and one part has value $w$, it follows that $\mathrm{GI_{abs}} \sim \frac{w}{n}$.

The *Theil index* is defined as

$$\mathrm{TI} = \frac{1}{n} \sum_{i=1}^{n} \frac{x_i}{\bar{x}} \cdot \log\left(\frac{x_i}{\bar{x}}\right), \tag{3}$$

where $x$ is an observed value (such as a specific incidence rate), $n$ is the number of values observed, $\bar{x}$ is the mean value and log stands for the natural logarithm. In a setting of n parts of a unit where $n − 1$ parts have value 0 and one part has value $w$, it follows that $\bar{x} = \frac{w}{n}$ and $\mathrm{TI} = \log(n)$. This motivates the fact that the TI has the maximum log(n), the largest inequality value between $n$ observations.

We use the bias-corrected bootstrap to calculate the confidence interval of the indices of interest [29, 30].

### Poisson regression for infection fatality rates

This section introduces classical testing theory to assess the null hypothesis *nIFR is equal over the parts of a unit*. To reject the null hypothesis we apply Poisson regression and the likelihood-ratio test. The dependent variable is the absolute cumulative death count per observational unit (i.e. federal state or country) and the independent variable is the state or country itself. We used absolute cumulative infection counts over a specific region and time period as offset. The likelihood ratio test is performed between the model containing the countries and an offset-adjusted intercept (M1) with the model without the countries (M0).

## Results

### Overview of the global data

Up to February 2, 2021, globally around 104 million (103,869,117) SARS-CoV-2 infections and 2.3 million (2,253,049) related deaths have been reported. The transmission of the virus has been documented in total of 190 countries and the reported number of infections per country varies between 1 (Micronesia, Vanuatu) and 26.3 million (U.S.). Deaths related to the virus have been documented in 179 countries and the number of deaths per affected country varies between 0 and 446,881. The three countries with the highest number of reported infections (deaths) are the U.S., India, and Brazil (the U.S., Brazil, and Mexico).

Table 1 shows an overview of the COVID-19 data for different continents, the EU and the U.S. Shown are number of states per region, the corresponding population size, cumulative

**Table 1. Descriptive overview of COVID-19 data for different geographic regions.**

| Region | No. of states | Population | Infections | Deaths | Naive IFR in % | p value LRT |
|--------|---------------|------------|------------|--------|----------------|-------------|
| Africa | 54 | 1,306,320,572 | 3,597,149 | 92,364 | 2.6 | <0.001 |
| America | 35 | 1,009,950,130 | 46,304,392 | 1,069,161 | 2.3 | <0.001 |
| Asia | 42 | 4,460,056,021 | 20,098,563 | 339,674 | 1.7 | <0.001 |
| Europe | 50 | 847,883,310 | 33,836,929 | 750,905 | 2.2 | <0.001 |
| EU | 27 | 446,824,564 | 19,873,376 | 479,020 | 2.4 | <0.001 |
| U.S. | 51 | 328,239,523 | 26,330,710 | 444,881 | 1.7 | <0.001 |

Overview of cumulative infections, deaths and naive infection fatality rate (IFR) calculated as the ratio of documented deaths to documented infections. The data are cumulated from the begin of the pandemic up to February 2, 2021. EU = European Union, U.S. = United States, LRT = likelihood-ratio test.

number of infections and infection related deaths from the start of the pandemic until February 2, 2021. A naive IFR estimate is calculated as the ratio of the number of documented deaths to number of documented infections. The table also provides a p-value to assess the null-hypothesis of a homogeneous nIFR within the specific geographic region.

In the following, we report inequality indices over specific regions during the first year of the pandemic. The main body of the paper contains the results for the relative and absolute Gini index. The supplement contains the results for the Theil index (see S1 and S2 Figs).

## Inequality in international COVID-19 data

$GI_{rel}$ and $GI_{abs}$ over time for the continents Africa (n = 54 countries), America (n = 35 countries), Asia (n = 42 countries) and Europe (n = 50 countries) are shown in Figs 1 and 2, respectively. The left column of Fig 1 shows the $GI_{rel}$ values for the documented parameters IR and MR of each continent. The $GI_{rel}$ values for Africa and Asia are above 0.6. The values for the Americas are a bit lower. Only in Europe, the $GI_{rel}$ of IR and MR decrease below 0.6. The $GI_{rel}$ for IR and MR look comparable for the A-continents (Africa, America and Asia). In Europe, there is higher inequality regarding MR compared to IR. In the right column of Fig 1 the $GI_{rel}$ for nIFR is shown. In Europe, nIFR $GI_{rel}$ values decrease over time while higher nIFR $GI_{rel}$ values are observed in the other continents. At the last time point $GI_{rel,nIFR} = 0.37$ (95% confidence interval CI: 0.30, 0.47) for Europe, $GI_{rel,nIFR} = 0.55$ (CI: 0.46, 0.66) for Africa, $GI_{rel,nIFR} = 0.44$ (CI: 0.35, 0.58) for America, and $GI_{rel,nIFR} = 0.61$ (CI: 0.50, 0.73) for Asia, respectively. Compared to Europe, a decrease of the nIFR $GI_{rel}$ values over time is less pronounced in the three A-continents.

A different view on inequality is given by the $GI_{abs}$ values presented in Fig 2, where inequality in absolute terms in shown. The six panels for the three A-continents (Fig 2a to 2f) show slight increases in $GI_{abs}$ values for IRs and MRs over time. The two panels for Europe (Fig 2g and 2h) reflect the effects of the two pandemic waves and the relaxed situation during the summer 2020. While the pandemic expands, absolute differences increase. From Eq (2) it follows that a comparison between the $GI_{abs,IR}$ and $GI_{abs,MR}$ gives a rough estimate of the corresponding geographic nIFR. For winter 2020, the nIFR in Europe is about 2% while the nIFR of Africa, the Americas and Asia are 2.5%, 2.5% and 1.5%, respectively. The $GI_{abs,nIFR}$ time course follows the time course of the relevant parameters IR and MR: The mean of absolute differences between observations correlates with the mean of these observations.

The results of the continental Theil indices are shown in S1 Fig. The TI time course follows qualitatively the $GI_{rel}$ time course. The maxima of TI values differ between continents, since each continent consists of a different number of countries. At the beginning, there is large inequality for both IR and MR. The time course of TI is similar for both IR and MR. Africa and Asia have the highest values, Europe the lowest, the Americas are in the middle position. Compared to the range of possible values, the indices are predominantly in the lower half or lower third. The TI values for Europe are even close to zero. The TI values for the nIFRs are less regular compared to those presented using the $GI_{rel}$.

## Inequality in U.S. and EU data

Fig 3 shows the time course of the incidence data for infections and mortality for the U.S. and the EU. They are used to calculate the inequality indices after being accumulated over a period of seven days.

Fig 3a and 3b show IR per 100,000 population for each of the 51 U.S. federal and of the 27 EU member states, respectively. Prominent peaks in the U.S. data (Fig 3a) can be seen for the states of New York in the spring and Arizona and Florida during the summer. The highest IR

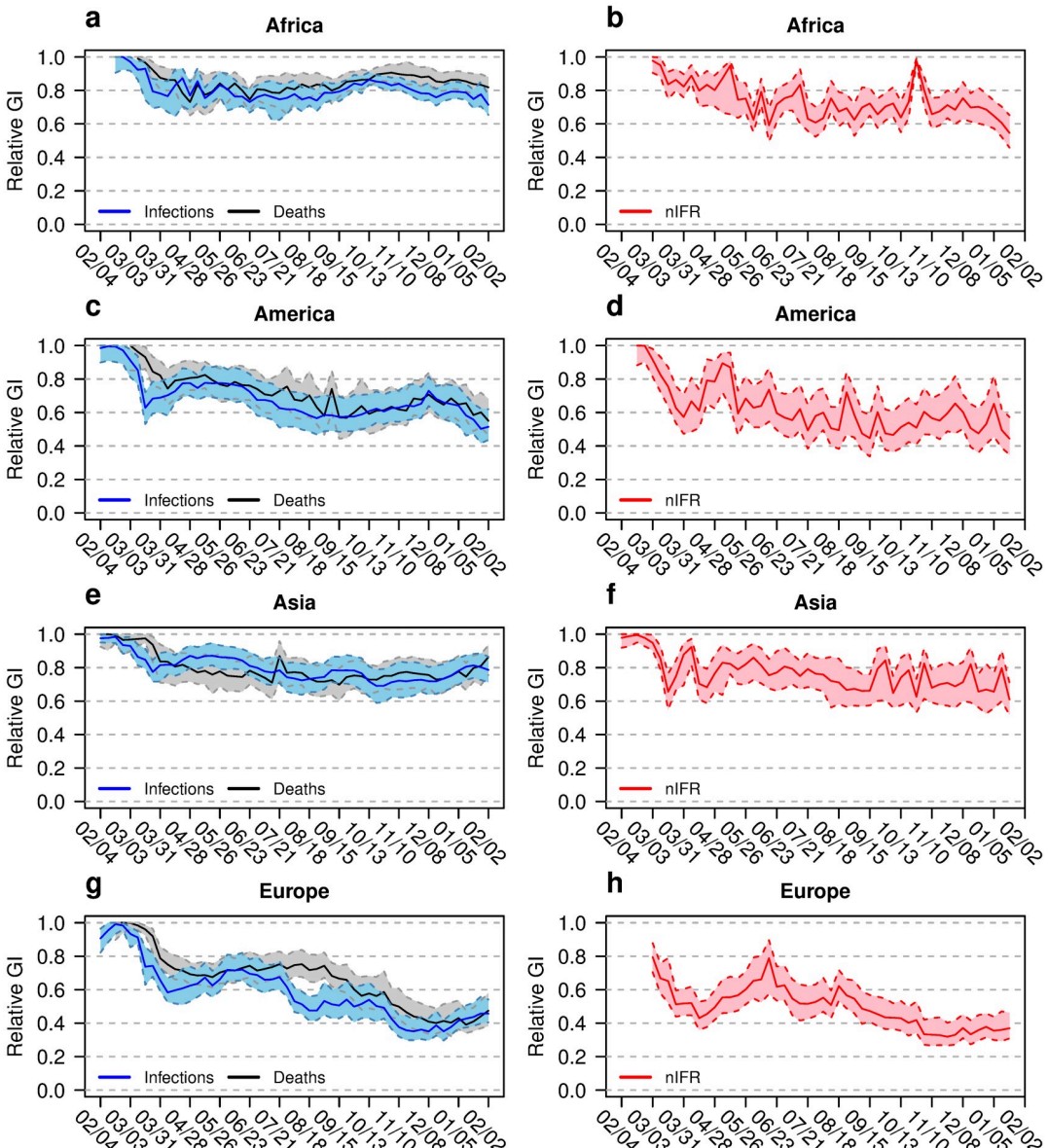

**Fig 1. Relative Gini indices for different continents.** Relative Gini indices (GI) for infection and death rates (left column) and naive infection fatality rate nIFR (right column) for Africa (a and b), America (c and d), Asia (e and f) and Europe (g and h). The coloured bands show the 95% confidence intervals. The horizontal axis shows the calendar time between February 2020 and February 2021.

of over 1,200 new infections per 100.000 population during one week is seen for North Dakota in the fall of 2020.

In the EU (Fig 3b), Luxembourg had the highest 7-day IR during the first wave in spring 2020. During the fall and winter Belgium, Lithuania, and France and Denmark show the highest IRs.

In terms of MR (Fig 3c), similar to IR, New York and New Jersey reported the highest MRs within the U.S. in the spring 2020. In the fall and winter, South Dakota and, more recently, Alabama reported the highest MRs. In the EU in spring 2020, the highest MRs were reported

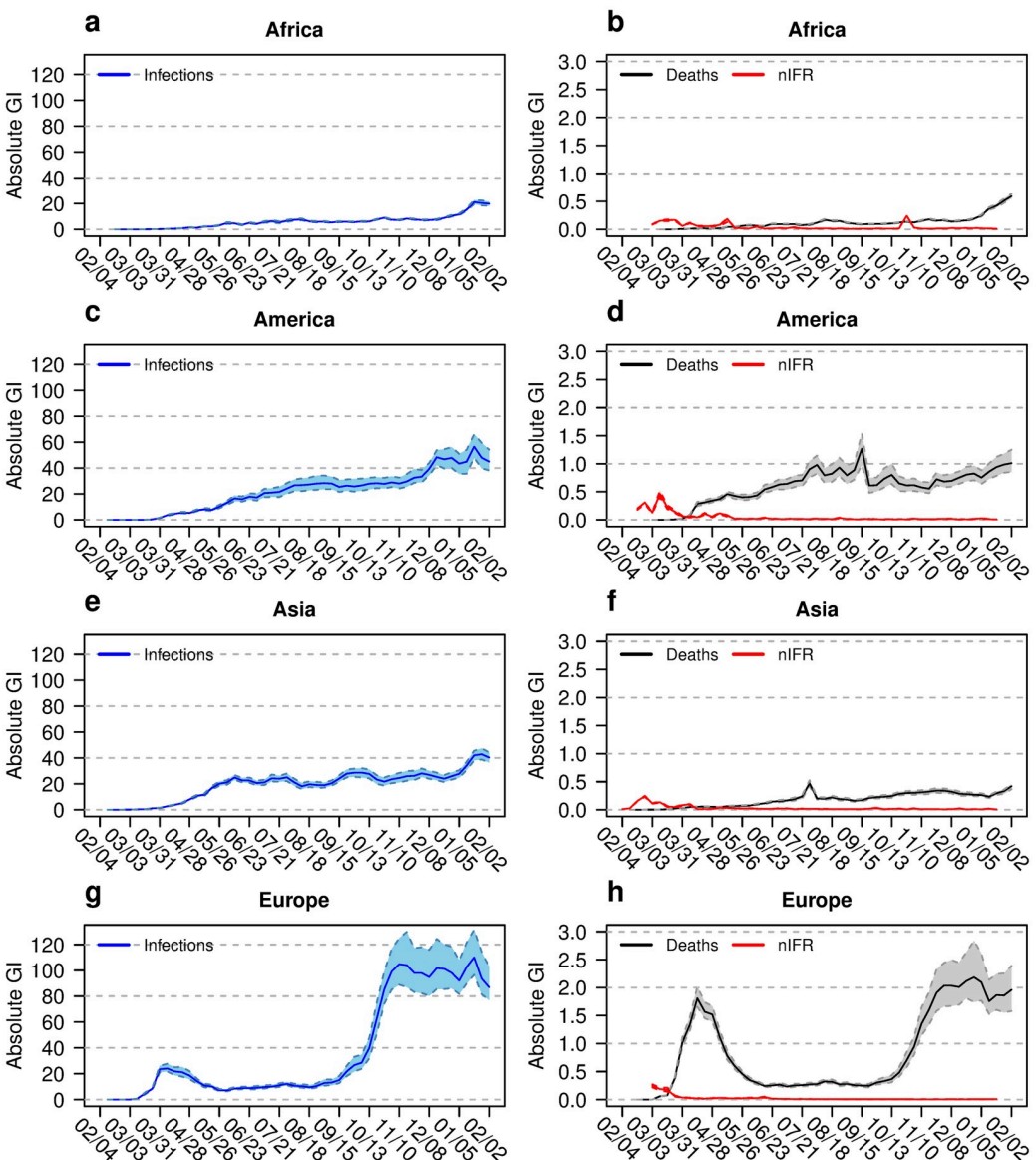

**Fig 2. Absolute Gini indices for different continents.** Absolute Gini indices (GI) for infection rates (left column), and for death rates and naive infection fatality rate nIFR (right column) for Africa (a and b), America (c and d), Asia (e and f) and Europe (g and h). The coloured bands show the 95% confidence intervals. The horizontal axis shows the time between February 2020 and February 2021.

for Belgium (Fig 3d). During the second wave in Europe, Czechia, Slovenia, and most recently Portugal reported the highest MRs within the EU member states. Fig 3e and 3f show the state-specific nIFRs, respectively.

The $GI_{rel}$ and $GI_{abs}$ values calculated from the incidence data of Fig 3 are shown in left column of Fig 4 for the U.S. and in the right column for the EU. A more detailed picture of the inequality in nIFR for both regions is shown in Fig 5.

The $GI_{rel}$ values for infections are below the $GI_{rel}$ values for mortality (Fig 4a and 4b). The time courses for the $GI_{rel}$ of both parameters decrease and represent more equality between federal or member states during the pandemic. The incidence curves represented in Fig 3

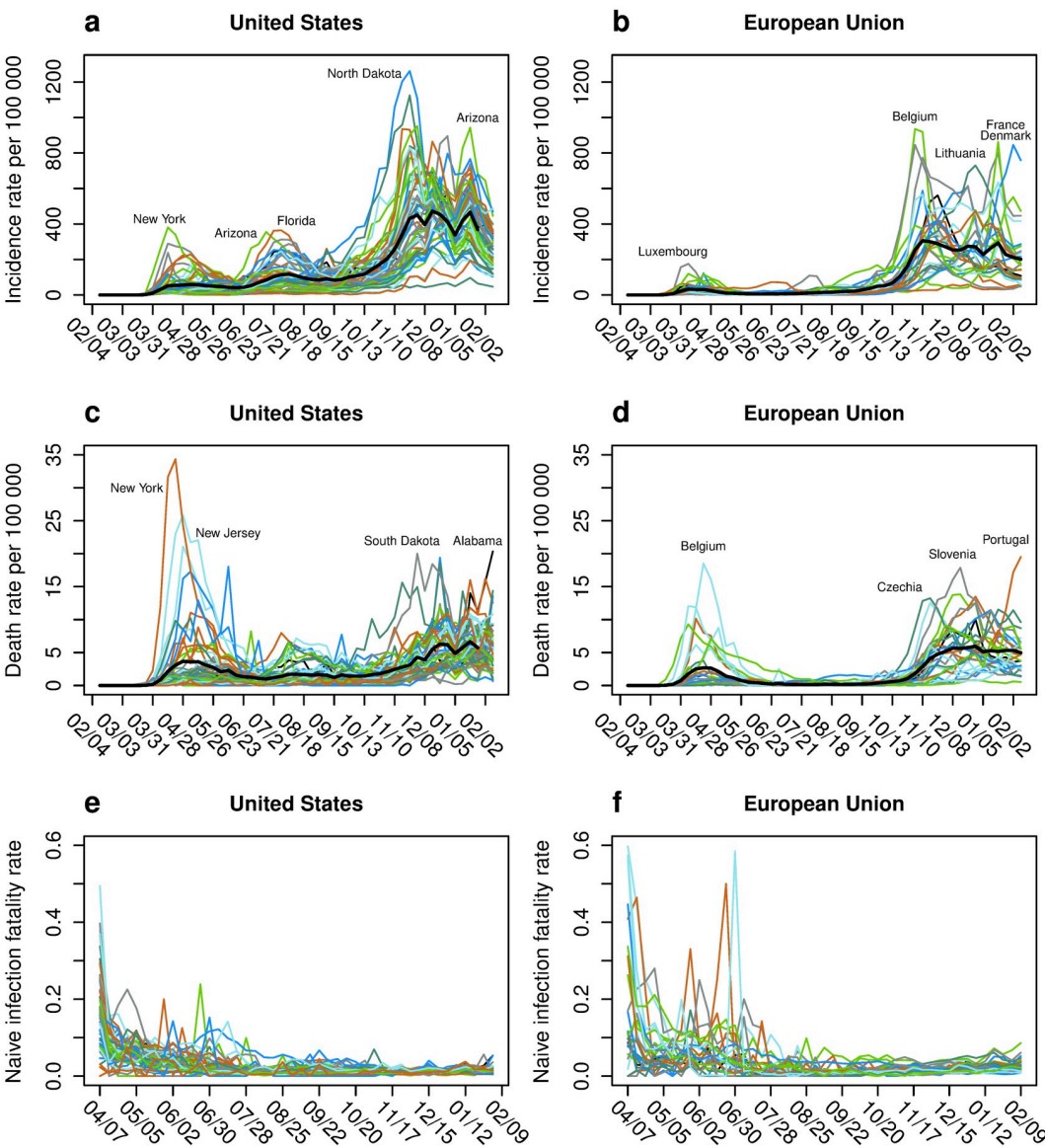

**Fig 3. Infection, death and infection fatality rates for the United States and the European Union.** Infection rate, death rate and naive infection fatality rate (nIFR) for the states of the United States (left column) and the member states of the European Union (right column). Panels a and b show the infection rate per 100,000 population over 7 days, panels c and d the death rate per 100 000 population over 7 days and panels e and f the time course of the nIFR. The horizontal axis shows the time between February 2020 and February 2021. The black thick lines show the mean incidence and death rate across all states.

demonstrate that for the U.S. (a and c) as well as for the EU (b and d) equality increases as the second wave of the pandemic begins. The time course of $GI_{rel,nIFR}$ values demonstrate similar index values during the winter 2020/2021. At the last observation time the $GI_{rel,nIFR} = 0.28$ (CI: 0.21, 0.37) for the U.S. and $GI_{rel,nIFR} = 0.34$ (CI: 0.27, 0.46) for the EU, respectively. The hump in the European $GI_{rel,nIFR}$ values during the summer 2020 is a result of low IR and MR values. Here two methodological aspects of the $GI_{rel}$ may be instrumental: Given a fixed mean absolute difference between observations, the $GI_{rel}$ decreases if the mean of the observations increases, the $GI_{rel}$ increases if the mean of the observations decreases.

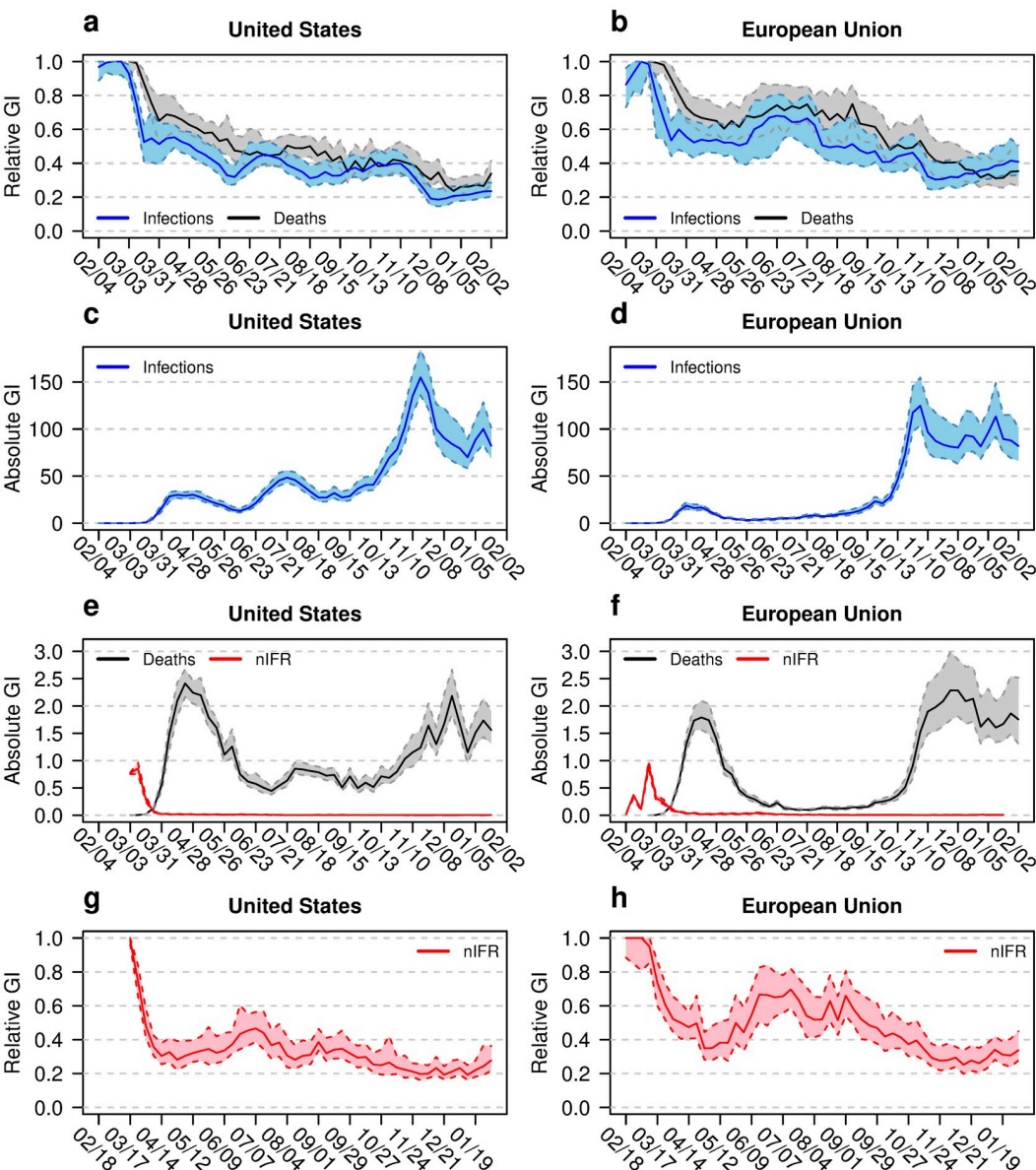

**Fig 4. Inequality measures for the United States and the European Union.** Gini indices (GI) for the states of the United states (left column) and the member states of the European Union (right column). Shown are the relative GIs of infections and deaths (blue and black curves, a and b), absolute GIs of infections (c and d), and absolute GIs of deaths and naive infection fatality rate nIFR (e and f) as well as relative GI for nIFR (g and h). The coloured bands show the 95% confidence intervals. The horizontal axis shows the time between February 2020 and February 2021.

The $GI_{abs}$ values reflect the seasonality of the pandemic: high $GI_{abs}$ values during the both waves, low value during the summer without excessive pandemic activity. While the EU shows higher MR inequality compared to the U.S. during the first wave of the pandemic, the second wave shows comparable values between the EU and the U.S. Based on the argument that the comparison between $GI_{abs,IR}$ and $GI_{abs,MR}$ reflects the nIFR, the time course of both indices allows a crude nIFR estimate over time. For winter 2020, the nIFR in the EU is about 2.5% while the nIFR in the U.S. around 1.8% (see also Table 1). The values of the $GI_{abs,nIFR}$ get very low with time (Fig 5c and 5d). In February 2021, the $GI_{abs,nIFR} = 0.004$ (CI: 0.003, 0.006) for

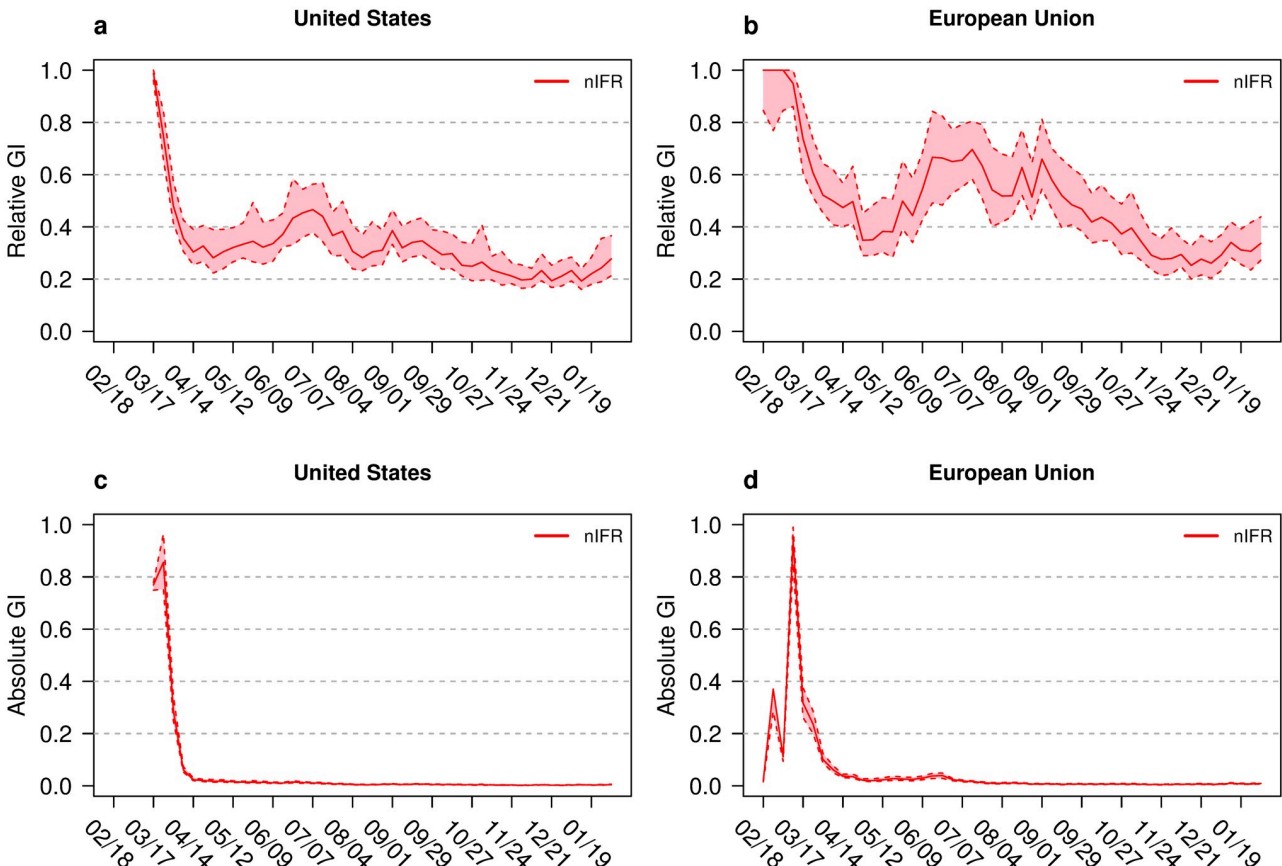

**Fig 5. Gini indices for naive infection fatality rate nIFR for theUnited States and the European Union.** Gini indices (GI) for the states of the United States (left column) and member states of the European Union (right column). Shown are the relative GIs of nIFR (a and b) and absolute GIs of nIFR (c and d). The coloured bands show the 95% confidence intervals. The horizontal axis shows the time between February 2020 and February 2021.

the U.S. and $GI_{abs,nIFR} = 0.009$ (CI: 0.007, 0.012) for the EU. These values are comparable to the nIFR values calculated based on cumulative data shown in S1 and S2 Tables.

The TI values are presented in S2 Fig. The left column shows TI values based on the U.S. data, the right column TI values based on the EU data. For all three parameters of interest (upper row: TIs for IR and MR; lower row: TIs for nIFR) the figures show a high level of inequality at the begin of the pandemic and a pronounced drop after the first wave to low TI values indicating equality. The dynamic of the TI values is comparable to that of the $GI_{rel}$ values. The visual impression (especially the sharp drop of values after the first wave) indicates higher equality in terms of TI compared to the $GI_{rel}$ value.

(Member) State-specific nIFRs for both regions can be found in the (S1 and S2 Tables).

## Discussion

During a pandemic, differences in incidence and number of deaths between countries and regions are not surprising findings. What does it add to use specific metrics to monitor the inequalities within larger political or geographic units over time? What are the political and/or moral implications? What does it mean for policy making?

As citizens of the EU, the authors miss a concise communication on the state of the European Union regarding central pandemic indicators. We consider metrics for equality/inequality as more comprehensive than looking at maps, following them over shorter or longer periods, performing an eye ball test by relating the gap between the best or worst performer, or discussing a series of caterpillar plots for the respective parameters. Also an U.S. American citizen wishes to be informed about how the pandemic load is distributed over his or her country. As the result of our paper, the $GI_{abs}$ provides an easy to understand quantitative statement which summarizes complex information.

The need for concise quantitative information on the pandemics time course over large geographical regions also holds for other global political structures in different continents to coordinate political interests (OAU—Organization of African Union; ASEAN—Association of Southeast Asian Nations; UNASUR—Union of South American Nations). The spread of the epidemic within these bodies is also of high political interest. As measures for the inequality of national income guide many aspects of policy making, measures for the inequality in health related issues are (especially over longer time periods of heavy dynamics) of similar importance. Policy making in a larger geographic area with high inequalities should build on local or regional measures. Policy making in a larger geographic area with high equality can build on more centralized components.

The causes of geographic inequality are complex. Inequality between nations reflects distinct health care systems, incompatible administrative structures, but also disparate documentation and reporting processes. Differences in demography will cause inequality regarding the disease measures of interest. Countries with young populations (like India with a mean age of 28 years) may show a different death rate compared to countries with higher average age. This paper does not study the features of nations that contribute to inequality, but focuses on how to transform inequality into accessible information.

The aim of this paper is to explore the relevance of well established indices to monitor inequality in SARS-CoV-2 infection and COVID-19 death incidences over time, taking continental, or national views. Three inequality indices are considered: The relative Gini index $GI_{rel}$, the absolute Gini index $GI_{abs}$, and the Theil index TI. They are widely used in the economic as well as in the health-economic literature. There are several pros and cons for each of these candidates.

The $GI_{rel}$ is the oldest and best known inequality index. Formally, the $GI_{rel}$ can be used to compare inequalities between larger groups of countries (between continents or between the U.S. and the EU). However, $GI_{rel}$ values have to be interpreted with care: Low $GI_{rel}$ (increasing equality, decreasing inequality) can be observed in settings where the pandemic aggravates and countries get more equal in a worse misery. Decreasing infection or mortality rates, on the other hand, can result in higher $GI_{rel}$ values indicating more inequality. The $GI_{rel}$ is equal to 1 in a setting of a well-contained epidemic where one country bears the entire burden of infection, while the remaining countries show no epidemic activities.

The $GI_{abs}$ does not show such behavior. A $GI_{abs,IR} = 100$ means that the the mean absolute difference in infection rates is 100 between two randomly selected countries. The actual range of the $GI_{abs}$ is not uniformly fixed as for the $GI_{rel}$, but depends on the specific setting. In our examples the ranges for $GI_{abs,IR}$ and $GI_{abs,MR}$ are quite different. The $GI_{abs}$ makes sense for comparisons over time within the same group. A crude nIFR estimate results by comparing the $GI_{abs,MR}$ with $GI_{abs,IR}$.

The Theil index uses the concept of entropy and its maximum depends on the group size. Therefore, it is also useful for comparisons over time. A high TI value can be understood as quantifying the evidence that a high rate comes from a particular country. This is the same as the share of a specific country in the distribution of the rates. A low TI value indicates a

situation where it is difficult to decide if a specific country has low or high values. This is a property also shared with the $GI_{rel}$. The TI and $GI_{rel}$ are sensitive to the relative distribution of low and high values within observations. Both show high values if only a few units are represented with large observations and both decrease as more units occupy larger values. The TI seems to be more liberal towards equality by showing more distinct drops towards smaller values during the pandemic.

We observe in our data that the $GI_{abs}$ rises while the $GI_{rel}$ decreases. This holds for infection as well as mortality rates. While the $GI_{rel}$ mirrors more equality, inequality in an absolute sense increases. The $GI_{rel}$ makes inequality issues regarding central pandemic indicators into a more salient political issue.

Of all continents Africa shows the highest relative and the lowest absolute inequality. This might be related to underreporting due to limited testing capacities and/or access to health care. Similar to Africa also Asia shows high relative and low absolute inequality. Asia contains a heterogeneous mixture of countries, so that it was surprising to find that low absolute inequalities: China which handled the epidemic apparently successful, India with a very young population and Japan with very old population. Japan has also a successful epidemic management with daily case counts far below the counts of Germany. The Americas as continent show steadily increasing absolute inequality in both infection and mortality rates. The European continent shows a very different shape of absolute inequality in infection and mortality rates over time, clearly reflecting both pandemic waves.

A specific interest of this study is to explore inequality between infection fatality rates across geographic areas over time. As a reviewer pointed out: "*Inequalities in IFR can be argued to be of moral significance (fairness), and such inequalities may also have policy implications as countries/regions should aim the lowest possible IFR.*" The results regarding the inequality in nIFR are presented in several figures: Panels b, d, f, h of Fig 1 show the relative inequality for the continents with Europe showing the lowest inequality. Panels b, d, f, h of Fig 2 show the absolute inequality. Since the $GI_{abs}$ is the mean absolute difference in the variable of interest between two randomly selected countries, the $GI_{abs,nIFR}$ has a direct interpretation as the mean difference in nIFR between two randomly selected countries. Panels e, f, g, h of Fig 4 show the relative and absolute inequality in nIFR for the U.S. and the EU. The relative inequality in the U.S. seems lower than for the EU during the summer 2020. In the winter the relative inequality evens out. The absolute inequality points towards mean nIFR difference of 0.4% (CI: 0.3%, 0.6%) between two random U.S. states and 0.9% (CI: 0.7%, 1.2%) between two random EU member states, respectively.

A comparison between the U.S. and the EU based on inequality indices is of high interest. There have been many rumours in the European media about how the U.S. performed during the pandemic. Further, the media coverage of the EU situation offered many narratives with permanently changing focus. Therefore, the paper takes a meta-look by using inequality indices for a broader comparison between both political bodies. Figs 4 and 5 show regional variability in the three pandemic key indicators. $GI_{rel}$ as well as $GI_{abs}$ drop down to low values reflecting equality in absolute as well as relative terms.

We see the quantification of inequality in COVID-19 key data across different geographic regions as a strength of our study. We considered three different inequality indices and discussed their strengths and weaknesses. Inequality was evaluated at different geographical scales (continental, national) and by means of Poisson regression. We see the concept of inequality as a complement to the dichotomous statistical thinking of difference yes/no. We also compared the inequality between the EU and the U.S. providing insights into the matter of relative and absolute inequality. We see it as important that mortality risk, infection risk, risk to be hospitalized or the type of treatment does not depend on where one lives. How close a

community is to achieving this goal is quantified by specific measures of inequality. This stresses the relevance of the presented results.

Our study also comes with some limitations, which are mainly related to the quality of the data we work with. Different countries might report differently. While some countries do not have a systematic reporting system, others might only report suspected COVID-19 or PCR-confirmed cases. Same thoughts can be applied to reporting of deaths, too. In addition to this, sudden sharp increases in infection and/or death counts might be attributed to late or irregular reporting and not point towards emerging situation of concern.

Due to the nature of reporting data we are unable to adjust the incidence and death rates for demographic features such as age or sex. It is well known that women and young persons have lower risk to die compared to men and people above 70 years [31]. Our analyses also do not account for seasonality or weather or the geographical location of the different areas. We also do not consider weighted inequality indices. We use the equal weighting approach per country because of our interest to assess the quality of the environment a person lives in and to compare environments. Introducing population weights would result in diminishing the influence of countries like the small European states in comparison to France or Germany, or in case of the U.S., or weighting the situation in states with large metropolitan areas in comparison to states with a more rural structure. Policy making is in general focused on countries and administrative/political units. Therefore, all countries/regions contribute equally to the calculation of inequality indices. We only adjust for the population size to report rates per 100,000 inhabitants in 7 day periods.

For the interpretation of our results, we have to additionally take into account that the number of infected persons is different from the number of positively tested persons. This is addressed by introducing the term *naive IFR (nIFR)*. Furthermore, the number of positively tested persons depends on the number of tests performed within a population. Therefore, the numbers used are biased estimates of the true incidences. Unfortunately, we do not have data on testing strategies that would allow us to adjust the numbers used in our article toward better estimates. Therefore, the true COVID-19 prevalence, incidence, and mortality remain uncertain. Rahmandad et al. discuss the impact of under-reporting across 92 nations on estimation of COVID-19 disease measures [32]. Their estimated cumulative cases and deaths through 7.0 and 1.4 times official reports, with substantially varying underreporting across the countries.

An important future application of inequality indices could be monitoring inequality in COVID-19 immunization rates between countries with early immunization activities (United Kingdom, Israel) and countries of the EU (as data become available).

## Conclusion

This paper applies three indices to measure inequality in international SARS-CoV-2 related infection and mortality data: the relative and absolute Gini index ($GI_{rel}$, $GI_{abs}$) as well as the Theil index (TI). These indices are applied to central pandemic indicators: infection (IR), mortality (MR), and (naive) infection fatality rate (nIFR). TI and $GI_{rel}$ show a comparable behavior. The $GI_{rel}$ and $GI_{abs}$ behave complementary: Often, the $GI_{rel}$ mirrors more equality, while absolute inequality as measured by $GI_{abs}$ increases. For this reason both indices should be reported simultaneously. Following the inequality indices over time allows a meta-look on very complex dynamics. Our analyses allow the conclusion that there is no convincing evidence that the U.S. and the EU show clear differences in specific health equity issues with respect to COVID-19 related mortality risks. Generally, the public discussion is directed towards the disease activity at a national or regional level. As it seems, only specialists are aware of the different aspects of the disease at a larger scale. For this reason we believe that

tools are needed to represent the pandemic state of a nation/region within a broader and easy to understand framework.

## Supporting information

**S1 Fig. Theil index for different continents.** Time course of Theil index for infection and death rates (left column) as well as for naive infection fatality rate (nIFR, right column) is shown for Africa (a and b), America (c and d), Asia (e and f) and Europe (g and h). The coloured bands show the 95% confidence intervals. The horizontal axis shows the time between February 2020 and February 2021. The black dashed horizontal line shows the maximum possible value of the Theil index.
(TIF)

**S2 Fig. Theil index for United States and European Union.** Time course of Theil index for infection and death rates (upper row) as well as for naive infection fatality rate (nIFR, lower row) is shown for the states of the United States (left column), and the member states of the European Union (right column). The coloured bands show the 95% confidence intervals. The horizontal axis shows the time between February 2020 and February 2021. The black dashed horizontal line shows the maximum possible value of the Theil index.
(TIF)

**S1 Table. Infection and death data for the European Union.** Cumulative infection and death rates per 100000 inhabitants, cumulative number of infections and deaths, and the naive infection fatality rate in % are shown for the 27 current member states of the European Union as of February 2, 2021.
(PDF)

**S2 Table. Infection and death data for the United States.** Cumulative infection and death rates per 100000 inhabitants, cumulative number of infections and deaths, and the naive infection fatality rate in % are shown for the U.S. states as of February 2, 2021.
(PDF)

## Author Contributions

**Conceptualization:** Ulrich Mansmann.

**Data curation:** Kirsi M. Manz.

**Formal analysis:** Kirsi M. Manz, Ulrich Mansmann.

**Methodology:** Ulrich Mansmann.

**Supervision:** Ulrich Mansmann.

**Visualization:** Kirsi M. Manz.

**Writing – original draft:** Kirsi M. Manz, Ulrich Mansmann.

**Writing – review & editing:** Kirsi M. Manz, Ulrich Mansmann.

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
