## [Decision Letter · Decision Letter 0]

20 Jan 2021

PONE-D-20-33079

In the face of the pandemic, are all equal? On the suitability of the Gini index to monitor time and geographic trends in incidence and death data during the SARS-CoV-2 pandemic

PLOS ONE

Dear Dr. Manz,

Thank you for submitting your manuscript to PLOS ONE. After careful consideration, we feel that it has merit but does not fully meet PLOS ONE’s publication criteria as it currently stands. Therefore, we invite you to submit a revised version of the manuscript that addresses the points raised during the review process.

We look forward to receiving your revised manuscript.

Kind regards,

Simone Lolli

Academic Editor

PLOS ONE

Journal Requirements:

2. We note that Figures 6 and 7 in your submission contain map images which may be copyrighted.

a. You may seek permission from the original copyright holder of Figures 6 and 7 to publish the content specifically under the CC BY 4.0 license. 

Additional Editor Comments:

Dear authors, please take into consideration all the comments and the issues raised by the reviewers.

Reviewers' comments:

Reviewer's Responses to Questions

**Comments to the Author**

1. Is the manuscript technically sound, and do the data support the conclusions?

Reviewer #1: Partly

Reviewer #2: Yes

2. Has the statistical analysis been performed appropriately and rigorously? 

Reviewer #1: Yes

Reviewer #2: Yes

3. Have the authors made all data underlying the findings in their manuscript fully available?

Reviewer #1: Yes

Reviewer #2: Yes

4. Is the manuscript presented in an intelligible fashion and written in standard English?

Reviewer #1: Yes

Reviewer #2: Yes

5. Review Comments to the Author

Reviewer #1: Dear authors,

Thank you for an interesting manuscript. It seems that you have used the best possible data for such a study, and the graphical presentation of your findings is really nice. However, I have some general comments, mainly addressing the choice of methodology and implications of your findings.

1. As I read your manuscript, my main concern is that I am not sure if you are applying an appropriate methodology and what the implications of your findings are. Differences in incidence and number of deaths between countries are not surprising findings, what does it add to analyze this using a Gini index? What are the moral implications? What are the policy implications? As you show in your manuscript, the interpretation of the Gini index is different for an infectious disease where high geographic inequality is desirable as this shows that the infection has not spread to all geographical areas. This makes inequality in COVID19 different from other health variables that previously have been studied using the Gini index. It seems a bit strange to analyze inequality in this manner (where the best outcome is high inequality), and I am not sure how this relates to fairness. As you discuss briefly in your manuscript, the difference between the observed inequality in incidence and deaths is explained by differences in the infection fatality rate (IFR). To me, this seems to be a highly relevant point to explore further (maybe even as the main topic of the paper), e.g. analyzing the inequality in IFR across geographic areas. Inequalities in IFR can be argued to be of moral significance (unfair), and such inequalities may also have policy implications as countries/regions should aim at the lowest possible IFR.

2. The Gini index is one of several inequality measures that could have been used for the analysis. Why do you find the Gini index to be the best summary measure for assessing inequality in incidence and mortality from COVID19? I think you should discuss and argue for why the Gini index is the best measure, as compared to other metrics that could have been used. Also, your introduction should contain some more information on findings from previous studies that have used the Gini index to assess geographic inequality.

3. A couple of comments on how you have applied the Gini index to the data:

- Why did you choose the relative Gini index and not the absolute Gini index? Should we not be concerned with both the absolute levels and the inequality? The absolute and relative approach to inequalities both have their strengths and weaknesses, so I think you should include a brief discussion of why you have chosen a relative measure of inequality.

- Your analyses are done by country or region, without weighting for population size. I am not convinced that this is the most relevant approach, in my opinion we are concerned about inequalities between individuals living in different areas, not inequalities between the areas per se. If this is the case, I think the Gini index should be adjusted for population size.

- The section about the Gini index’ scale invariance in the method part is confusing. The Gini index for infection will be equal to the Gini index of deaths if, and only if, the IFR is constant across all regions (which it is not).

4. The manuscript is in general quite well written, but I think it could benefit from being a bit shorter and more concise, and additional language reviewing is still needed. Some parts need to be restructured, e.g. in the abstract the results should be presented before being discussed.

Good luck revising your manuscript!

Reviewer #2: The authors presented an interesting study to assess inequalities in COVID-19 transmission and deaths using as metric the Gini Index. The manuscript is generally well written and clear.

Major Issues:

Why the authors chose this index among others? Is there any specific characteristic of the index that makes it unique?

As stated by the authors, the different European states (or US states) adopt heterogenous policies to determine the effective number of deaths or infections. For this reason, is the Gini Index reliable if the infection and deaths are differently accounted?

it seems that the analysis for Bavaria and districts show a different outcome with respect to the other results.

An important factor that it is not mentioned it is the latitude dependency. As reported in literature, the different meteorological conditions impact on COVID-19 pandemic transmission. For this reason, instead of testing age and sex dependency, I would test latitude dependency.

Specific comments can be found in the attached file.

6. PLOS authors have the option to publish the peer review history of their article (what does this mean?). If published, this will include your full peer review and any attached files.

Reviewer #1: **Yes: **Eirin Krüger Skaftun

Reviewer #2: No

---

## [Author Response · Author response to Decision Letter 0]

26 Mar 2021

Figures 6 and 7 with potential copyright issues have been removed from the revised version of the manuscript. A point-by-point response to reviewers' comments is included as a part of the submission.

---

## [Decision Letter · Decision Letter 1]

26 Apr 2021

Inequality indices to monitor geographic differences in incidence, mortality and fatality rates over time during the COVID-19 pandemic.

PONE-D-20-33079R1

Dear Dr. Manz,

We’re pleased to inform you that your manuscript has been judged scientifically suitable for publication and will be formally accepted for publication once it meets all outstanding technical requirements.

Kind regards,

Simone Lolli

Academic Editor

PLOS ONE

Additional Editor Comments (optional):

I am happy that the manuscript is now ready for publication

Reviewers' comments:

Reviewer's Responses to Questions

**Comments to the Author**

1. If the authors have adequately addressed your comments raised in a previous round of review and you feel that this manuscript is now acceptable for publication, you may indicate that here to bypass the “Comments to the Author” section, enter your conflict of interest statement in the “Confidential to Editor” section, and submit your "Accept" recommendation.

Reviewer #2: All comments have been addressed

2. Is the manuscript technically sound, and do the data support the conclusions?

Reviewer #2: Yes

3. Has the statistical analysis been performed appropriately and rigorously? 

Reviewer #2: Yes

4. Have the authors made all data underlying the findings in their manuscript fully available?

Reviewer #2: (No Response)

5. Is the manuscript presented in an intelligible fashion and written in standard English?

Reviewer #2: Yes

6. Review Comments to the Author

Reviewer #2: I am happy that all my previously raised issue were properly addressed. Now the manuscript is ready to published after minor technical corrections.

7. PLOS authors have the option to publish the peer review history of their article (what does this mean?). If published, this will include your full peer review and any attached files.

Reviewer #2: No

---

## [Editor Report · Acceptance letter]

28 Apr 2021

PONE-D-20-33079R1 

Inequality indices to monitor geographic differences in incidence, mortality and fatality rates over time during the COVID-19 pandemic. 

Dear Dr. Manz:

I'm pleased to inform you that your manuscript has been deemed suitable for publication in PLOS ONE. Congratulations! Your manuscript is now with our production department. 

Kind regards, 

on behalf of

Dr. Simone Lolli 

Academic Editor

PLOS ONE